# Unexpected Crosslinking Effects of a Human Thyroid Stimulating Monoclonal Autoantibody, M22, with IGF1 on Adipogenesis in 3T3L-1 Cells

**DOI:** 10.3390/ijms24021110

**Published:** 2023-01-06

**Authors:** Araya Umetsu, Tatsuya Sato, Megumi Watanabe, Yosuke Ida, Masato Furuhashi, Yuri Tsugeno, Hiroshi Ohguro

**Affiliations:** 1Department of Ophthalmology, School of Medicine, Sapporo Medical University, Sapporo 060-8556, Japan; 2Department of Cardiovascular, Renal and Metabolic Medicine, Sapporo Medical University, Sapporo 060-8556, Japan; 3Department of Cellular Physiology and Signal Transduction, Sapporo Medical University, Sapporo 060-8556, Japan

**Keywords:** 3T3-L1 cell, IGF1, M22, three-dimensional (3D) tissue culture

## Abstract

To study the effects of the crosslinking of IGF1 and/or the human thyroid-stimulating monoclonal autoantibody (TSmAb), M22 on mouse adipocytes, two- and three-dimensional (2D or 3D) cultures of 3T3-L1 cells were prepared. Each sample was then subjected to the following analyses: (1) lipid staining, (2) a real-time cellular metabolic analysis, (3) analysis of the mRNA expression of adipogenesis-related genes and extracellular matrix (ECM) molecules including collagen (Col) 1, 4 and 6, and fibronectin (Fn), and (4) measurement of the size and physical properties of the 3D spheroids with a micro-squeezer. Upon adipogenic differentiation (DIF+), lipid staining and the mRNA expression of adipogenesis-related genes in the 2D- or 3D-cultured 3T3-L1 cells substantially increased. On adding IGF1 but not M22 to DIF+ cells, a significant enhancement in lipid staining and gene expressions of adipogenesis-related genes was detected in the 2D-cultured 3T3-L1 cells, although some simultaneous suppression or enhancement effects by IGF1 and M22 against lipid staining or *Fabp4* expression, respectively, were detected in the 3D 3T3-L1 spheroids. Real-time metabolic analyses indicated that monotherapy with IGF1 or M22 shifted cellular metabolism toward energetic states in the 2D 3T3-L1 cells upon DIF+, although no significant metabolic changes were induced by DIF+ alone in 2D cultures. In addition, some synergistical effects on cellular metabolism by IGF1 and M22 were also observed in the 2D 3T3-L1 cells as well as in cultured non-Graves’ orbitopathy-related human orbital fibroblasts (n-HOFs), but not in Graves’ orbitopathy-related HOFs (GHOFs). In terms of the physical properties of the 3D 3T3-L1 spheroids, (1) their sizes significantly increased upon DIF+, and this increase was significantly enhanced by the presence of both IGF1 and M22 despite downsizing by monotreatment, and (2) their stiffness increased substantially, and no significant effects by IGF-1 and/or M22 were observed. Regarding the expression of ECM molecules, (1) upon DIF+, significant downregulation or upregulation of Col1 and Fn (3D), or Col4 and 6 (2D and 3D) were observed, and (2) in the presence of IGF-1 and/or M22, the mRNA expression of Col4 was significantly downregulated by M22 (2D and 3D), but the expression of Col1 was modulated in different manners by monotreatment (upregulation) or the combined treatment (downregulation) (3D). These collective data suggest that the human-specific TSmAb M22 induced some unexpected simultaneous crosslinking effects with IGF-1 with respect to the adipogenesis of 2D-cultured 3T3-L1 cells and the physical properties of 3D 3T3-L1 spheroids.

## 1. Introduction

Graves’ orbitopathy (GO) is an autoimmune inflammatory disorder of orbital fatty tissue/connective tissue and extraocular muscles. GO is associated with several characteristic clinical manifestations, including upper eyelid retraction, edema, and erythema of the periorbital tissues and conjunctivae, as well as exophthalmos [1,2]. As a possible explanation of the molecular pathology of GO, it is known that autoimmune responses to the thyroid stimulating hormone receptor (TSHR) induce inflammation of orbital fatty tissues, resulting in an expanded volume of orbital adipose tissue, extracellular matrixes (ECMs) as well as extraocular muscles due to new fat cell growth [3,4]. In fact, the human thyroid stimulating monoclonal autoantibody (TSmAb), referred to as “M22”, isolated from the peripheral blood lymphocytes of a patient with GO [5], was reported to increase the expression of IL-6 in orbital preadipocyte fibroblasts and the secretion of IL-6 by mature adipocytes [6]. This finding suggests that circulating TSHR autoantibodies in GO might indeed play a direct role in the clinical manifestations associated with the pathogenesis of GO. Furthermore, based upon the well-known cross talk mechanism between G protein-coupled receptors (GPCRs) and receptor tyrosine kinases [7,8], it has been shown that a GPCR family member, TSHR, is also linked with receptor tyrosine kinases, i.e., the IGF-1 receptor (IGF-1R), in such cross talk mechanisms [7,8]. In fact, interactions between TSHR and IGF-1R signaling were reported to occur in primary cultures of GO-related human orbital fibroblasts (GHOFs) [9,10,11], and this signaling synergistically enhanced the secretion of hyaluronic acid (HA), a major pathogenic product of GO [2]. Based upon these collective findings, autoantibodies that bind to and stimulate TSHR [12], such as M22, and to IGF-1 to stimulate IGF-1R [13,14] have been used in studies of the pathogenesis of GO.

TSHR is a unique glycoprotein hormone receptor that is made up of A and B subunits that are linked by disulfide bonds. The A and B subunits are released by proteolytic cleavage. Among these two subunits, the A subunit on the cell surface is the primary immunogen and causes the production of thyroid-stimulating autoantibodies (TSAbs) such as M22 [15]. The three-dimensional structure of M22 complexed with the A unit of TSHR has been characterized by X-ray crystallography [16]. Although the crystallographic data clearly confirm that M22 is bound to the TSHR through the leucine-rich repeat (LRR) domain, the precise mechanism responsible for the M22-induced activation of TSHR causing GO remains to be elucidated. Regarding this issue, previous studies demonstrated that the binding properties of M22 with the TSHR A subunit were comparable to a nonstimulatory mouse monoclonal antibody (mAb) (referred to as 3BD10) and found some similarities as well as some differences between them [16,17,18,19]. Therefore, these results suggest that the human-specific TSmAb M22 may also have some effects on the mouse TSHR in addition to the human TSHR. In fact, and interestingly, Neumann et al. reported that serum free T4 levels were significantly increased in mice that were given M22 [20]. Therefore, based upon these collective findings, examining the effects of M22 and/or IGF1 on mouse-derived adipocytes would be of considerable interest because such simpler experimental conditions may facilitate a better understanding of the currently unknown mechanisms responsible for causing the M22-induced stimulation of TSHR.

In terms of physiology, the spreading and differentiation of adipocytes should occur within a three-dimensional (3D) space and not on a planar culture. In adipocyte-related research, the use of a 3D cell culture system would be more desirable compared to a two-dimensional (2D) planar cell culture method. Although the use of such 3D cell culture systems has not become popular because of technical difficulties associated with this methodology, the use of such cultures has recently emerged as a useful strategy for modeling several human diseases [21]. In fact, it has been proposed that such a 3D cell culture technique would allow for more representative physiological cell-cell and cell-extracellular matrix (ECM) interactions than conventional 2D culture models [22]. Quite recently, our group independently succeeded in producing 3D spheroids using 3T3-L1 cells [23,24,25,26,27,28] as well as cells from other sources, including human orbital fibroblasts (HOFs) [29,30,31], human corneal stromal fibroblasts [32], human trabecular meshwork (HTM) [33,34,35,36] and human conjunctival fibroblasts (HconF) [37,38,39,40].

Therefore, in this study, we report on an evaluation of the effects of M22 and/or IGF1 on several biological aspects of 2D- and 3D-cultured 3T3-L1 preadipocytes during their adipogenesis.

## 2. Results

It was previously reported that the anti-human TSHR antibody (referred to as M22) induced an increase in serum free T4 levels in mice [20], suggesting the presence of some interesting unidentified mechanisms effects beyond the species level. To determine whether such unexpected and unidentified cross-reactivities of M22 exist, especially with IGF1 toward mouse adipose tissue, we investigated the effects of IGF1 and/or M22 on the adipogenesis of 3T3-L1 preadipocytes, in which TSHR is not expressed (Appendix A). For this purpose, we employed 3D spheroid cultures of these cells in addition to the conventional 2D planar cell cultures. Upon adipogenic differentiation (DIF+), lipid staining and the mRNA expression of *Pparγ* and *Fabp4* in the 2D- or 3D-cultured 3T3-L1 cells substantially increased, as reported in our previous studies [24,25,26] (Figure 1 and Figure 2). However, the additive effects of IGF1 and/or M22 on DIF+ were different between the 2D and 3D cultures. Specifically, IGF1 but not M22 caused a significant enhancement in Oil Red O staining intensities and the gene expressions of *Pparγ* and *Fabp4* in the 2D-cultured 3T3-L1 cells (Figure 1). Such IGF1-induced increase effects were not observed by BODIPY lipid staining nor by the mRNA expression of these adipogenesis-related genes in the case of the 3D 3T3-L1 spheroids (Figure 2). However, some simultaneous enhancement effects by IGF1 and M22 toward lipid staining or *Pparγ* and *Fabp4* expressions were observed in the case of the 3D 3T3-L1 spheroids (Figure 2).

To further study the additive effects of IGF1 and/or M22 to DIF+, a real-time cellular metabolic analysis of the 2D 3T3-L1 cells was conducted using a Seahorse Bioanalyzer, and physical property measurements including the size and stiffness of the 3D 3T3-L1 spheroid were made. As shown in Figure 3, both OCR and ECAR were increased by a monotreatment of IGF1 or M22 in the 2D 3T3L-1 cells upon DIF+ despite the fact that no significant changes in OCR and ECAR were induced by DIF+ alone in the 2D-cultured cells. Quite interestingly, some IGF1- and M22-induced energetic effects on metabolism were synergistic. Similarly, such synergistic effects by IGF1 and M22 on cellular metabolism were also observed in TSHR-positive non-GO-related human orbital fibroblasts (n-HOFs), but not in GO-related HOFs (GHOFs), which may have already been exposed to IGF1 and M22 (Appendix A). These results suggest that the presence of IGF1 and/or M22, which mimic the pathogenesis of GO, may be partially synergistic, causing cellular metabolism to shift to a more energetic state, even independent of TSHR. Such unexpected synergistical effects were also observed in the physical properties of the 3D 3T3-L1 spheroids (Figure 4) as follows: (1) their size and stiffness were significantly increased and decreased upon DIF+ as was demonstrated in our previous reports [24,25,26], (2) the 3D spheroid size in DIF+ cells was significantly increased by the presence of both IGF1 and M22 despite downsizing by their monotreatment, and (3) no significant changes brought on by IGF-1 and/or M22 were observed in their stiffness.

To further study these diverse effects generated by IGF-1 and/or M22 between 2D and 3D cell culture systems, we evaluated the mRNA expressions of the major ECM proteins, COL1, 4 and 6, and FN. As shown in Figure 5, upon DIF+, significant downregulation or upregulation of *Col1* and *Fn* (3D), or *Col4* and *6* (2D and 3D) were observed, consistent with our previous studies [24,25,26]. In terms of the effects of IGF-1 and/or M22, the mRNA expression of *Col4* was significantly downregulated by M22 (2D and 3D) but that of *Col1* was differentially regulated by monotreatment (upregulation) or combined treatment (downregulation) (3D) (Figure 5). Such differences in the COL1 and COL4 expressions among treatment groups were also confirmed by immunolabeling of 2D- and 3D-cultured 3T3-L1 cells (Figure 6). Taking these collective data into account, we speculated that some unexpected, simultaneous, TSHR-independent effects of IGF-1 and M22 may be exerted and that these could be more clearly detected in the case of the 3D spheroid culture system.

Therefore, these collective results suggest that IGF1 and M22 may synergistically affect adipogenesis of 3T3-L1 cells in a TSHR-independent manner.

## 3. Discussion

It is known that M22 is a potent stimulatory human TSmAb that competitively binds to the TSH against the concave surface of the LRR domain, resulting substantial TSH-binding inhibiting activity [5,41]. Alternatively, M22 could also enhance the DIF+ of 2D cultured n-HOFs [42], similar to the case where TSH stimulates the conversion of mouse embryonic stem cells into adipocytes as a pro-adipogenic factor [43], since it is known that a significant increase in lipid accumulation occurs following the TSH-stimulated production of cAMP [44,45]. Interestingly, as with the human-specific TSmAb M22, it was also reported that KSAB1, the mouse-specific TSmAb, also exerted significant stimulating effects toward GHOFs despite the fact that there were differences in the response curve between both TSmAbs [46], suggesting the possibility that cross-reactivity could occur between M22 and mouse TSmAbs. In fact, it was reported that mouse TSmAbs inhibit the binding of 125I-TSH and 125I-M22 Fab to the human TSHR, although the inhibitory effects of the mouse TSmAbs were less than those of M22 [41]. In addition, both M22 and the murine TSAbs KSAB1 and KSAB2 also bind recombinant porins of Y. enterocolitica, which was identified as an infectious pathogen associated with GO [47,48,49], suggesting the possibility of cross-reactivity [50].

In addition to thyrocytes, the expression of TSHR has been detected in numerous extrathyroidal tissues, including the liver [51] and adipose tissues [52,53]. Concerning the role of extrathyroidal TSHR, it was shown that TSHR is an important regulator of adipocyte differentiation based upon the fact that the TSHR expression is increased during the adipogenic differentiation of TSHR-positive 3T3-L1 preadipocytes, and the knockdown of TSHR blocked this adipocyte differentiation [54]. Similar results were also reported for rat preadipocytes [52], n-HOFs [55,56] and mouse embryonic stem cells [43]. In the current study, we also observed that the human-specific TSmAb M22 induced some simultaneous crosslinking effects with IGF-1 toward adipogenesis in 2D cultures of 3T3-L1 cells and toward the physical properties of the 3D 3T3-L1 spheroids despite the absence of TSHR within the 3T3-L1 cells, suggesting the possibility that TSHR exerts effects that are independent of M22. Although the possible molecular mechanisms responsible for these TSHR-independent effects mediated by M22 on 3T3-L1 cells have not been elucidated at the time of writing, we speculate that some LRR [57]-related mechanisms may be involved. In fact, the LRR appears to have been identified within the primary structures of many proteins of diverse origins, including tyrosine kinase receptors [58], enzymes [59], cell adhesion molecules [60], and extracellular matrix-binding glycoproteins [61], in addition to the GPCR receptors for LH, FSH, and TSH [62,63,64,65,66,67,68]. To support this hypothesis, a previous study demonstrated that flightless-I (FLII), a transcriptional modulator of PPARγ, binds directly to PPARγ through the LRR domain and suppresses the transcriptional activity of cells [69]. Furthermore, in addition to the current observations related to the Seahorse real-time cellular metabolic analyses of n-HOFs and GHOFs (Appendix A), we previously reported that a significant difference existed between GHOFs and n-HOFs that had been treated with M22 and IGF1 in terms of mRNA expression levels of several molecules including ECM molecules (COL1, 4 and 6 and FN), ECM regulatory factors (LOX and CTGF), inflammatory cytokines (IL1B and IL6) and ER stress-related factors, suggesting the presence of some additional, unrelated TSHR effects by M22 and IGF1 [30]. However, this TSHR-independent cross-reactivity by M22 and IGF1 is speculative at this time, and additional studies such as a TSHR knockdown system and others will be required to reveal the magnitude of this issue.

## 4. Materials and Methods

Two-dimensional and three-dimensional cell cultures of 3T3-L1 cells, non-GO-related human orbital fibroblasts (n-HOFs) and GO-related HOFs (GHOFs) and adipogenic differentiation of 3T3-L1 cells with or without IGF1 and M22 were used.

In the presence or absence of 100 ng/mL IGF1 and/or 10 ng/mL M22, two-dimensional (2D) planar and three-dimensional (3D) spheroid cultures of 3T3-L1 preadipocytes (#EC86052701-G0, KAK) were maintained for 7 days [24,25], or n-HOFs [30,70,71] or GHOFs [29,72] were maintained for 10 days, as described previously. The induction of adipogenic differentiation of the 3T3-L1 cells was also carried out as described previously [24,25]. Lipid staining of the 2D- and 3D-cultured 3T3-L1 cells by Oil Red O and BODIPY, respectively, was performed as described previously [24,25]. (Information concerning the methods used in this study are shown in the Appendix A).

### 4.1. Measurement of Real-Time Cellular Metabolic Functions of 2D-Cultured 3T3-L1 Cells

The rates of oxygen consumption (OCR) and extracellular acidification (ECAR) of 2D 3T3-L1 cells were measured using a Seahorse XFe96 Bioanalyzer (Agilent Technologies) as described previously with minor modifications [73,74]. (Information concerning the methods used in this study are shown in the Appendix A).

### 4.2. Characterization of the 3D 3T3-L1 Spheroids

For the characterization of the 3D 3T3-L1 spheroids, their physical properties, the mean sizes (μm^2^) and stiffness (μN/μm) were measured as described previously [29]. (Information concerning the methods used in this study are shown in the Appendix A).

### 4.3. Other Analytical Methods

Immunostaining using antibodies against ECM molecules including collagen (COL1), 4, and 6, and fibronectin (FN), quantitative PCR using predesigned specific primers (Appendix A), and statistical analyses using Graph Pad Prism 8 (GraphPad Software, San Diego, CA) were performed as described previously [24,25]. All data are presented as arithmetic means ± the standard error of the mean (SEM) and statistical significance among experimental groups was evaluated by ANOVA followed by a Tukey’s multiple comparison test. (Information concerning the methods used in this study are shown in the Appendix A).

## Figures and Tables

**Figure 1 ijms-24-01110-f001:**
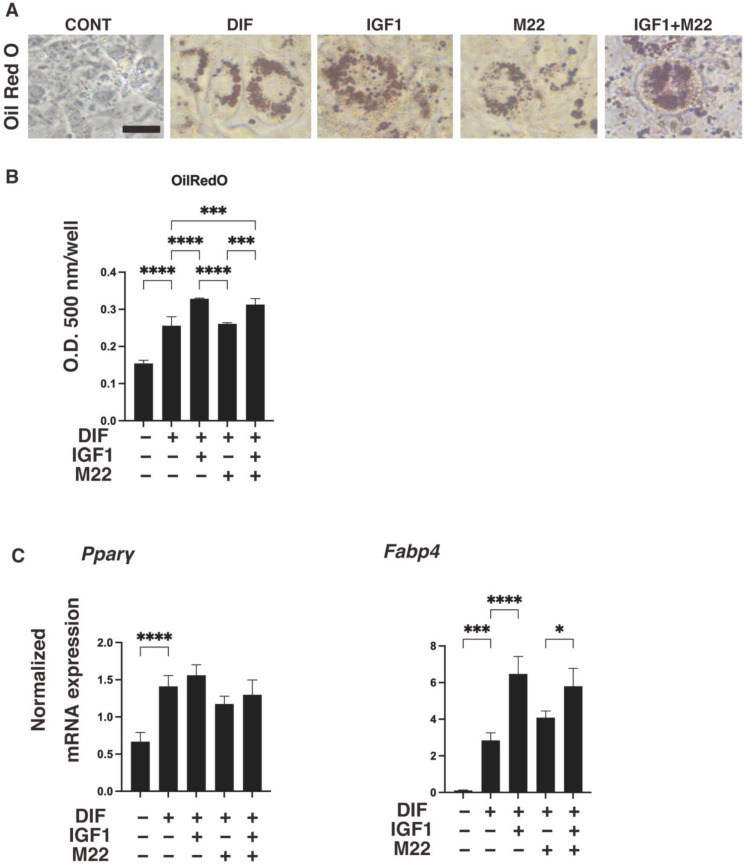
Effects of IGF1 and/or M22 on adipogenesis of 2D-cultured 3T3-L1 cells. Two-dimensional cultures of 3T3-L1 cells were prepared under several sets of conditions: preadipocytes of 3T3-L1 cells (DIF-) or adipogenic-differentiated preadipocytes (DIF+) with or without 100 ng/mL IGF1 and/or 10 ng/mL M22. The specimens were subjected to analysis by Oil Red O lipid staining (panel (**A**); representative phase contrast images, scale bar: 100 μm, and panel (**B**); their staining intensities, O.D. 500 nm) and qPCR of the adipogenesis genes *Pparγ* and *Fabp4* (panel (**C**)). All experiments were performed in duplicate using fresh preparations (total *n* = 4), each of which consisted of a confluent 12-well dish. * *p* < 0.05, *** *p* < 0.005, **** *p* < 0.001.

**Figure 2 ijms-24-01110-f002:**
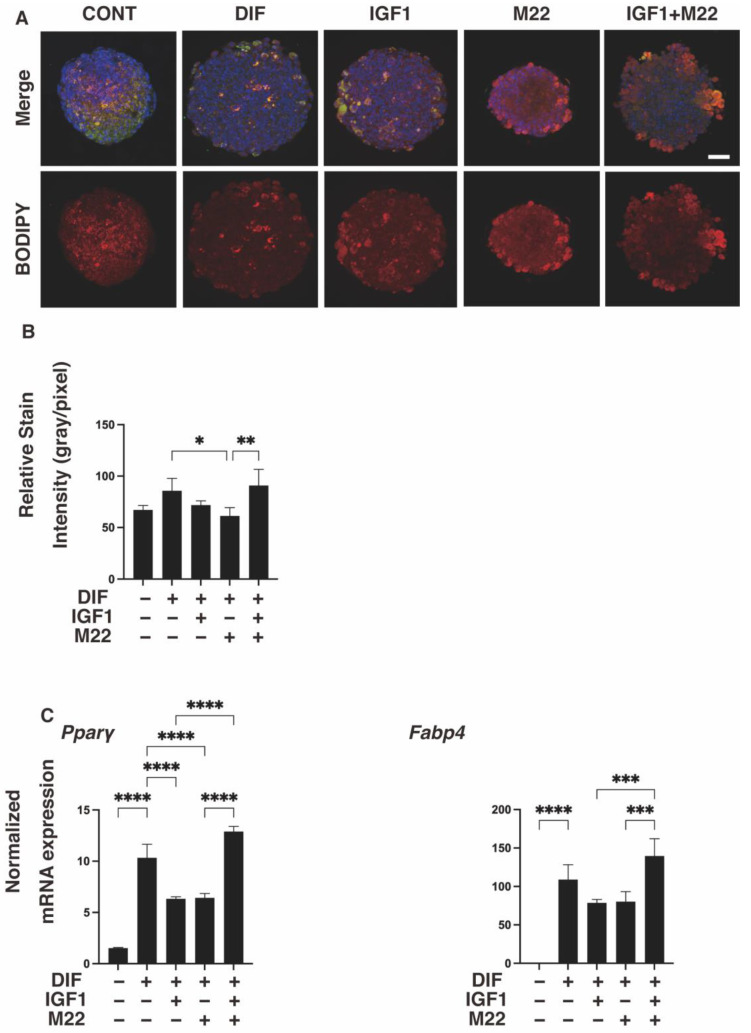
Effects of IGF1 and/or M22 on the adipogenesis of 3T3-L1 3D spheroids. Three-dimensional spheroids of 3T3-L1 cells were prepared under several sets of conditions: preadipocytes of 3T3-L1 cells (DIF-) or adipogenic-differentiated preadipocytes (DIF+) with or without 100 ng/mL IGF1 and/or 10 ng mL M22. They were immunostained with DAPI (blue), phalloidin (green) and BODIPY (red). Merged images and BODIPY images are shown in panel (**A**) (scale bar: 100 μm) and their staining intensities (gray/pixel) were plotted (panel (**B**)). The mRNA expression levels of adipogenesis-related genes, including *Pparγ* and *Fabp4*, under the above conditions were plotted in panel (**C**). All experiments were performed in duplicate using fresh preparations (total *n* = 4), each of which consisted of 16 spheroids. * *p* < 0.05, ** *p* < 0.01 *** *p* < 0.005, **** *p* < 0.001.

**Figure 3 ijms-24-01110-f003:**
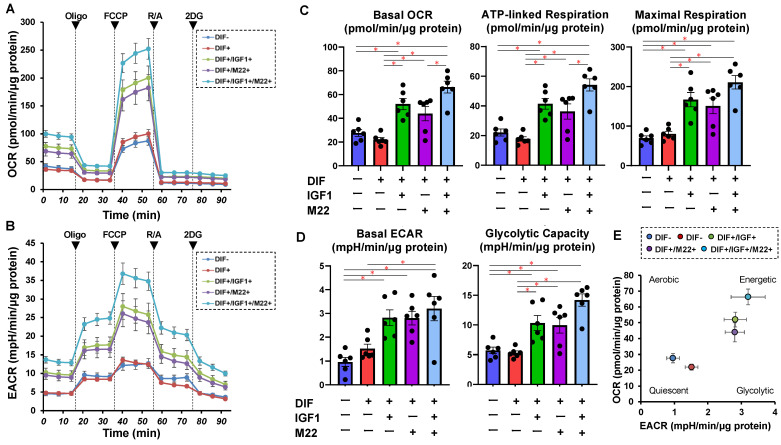
Effects of IGF1 and/or M22 on cellular metabolism in 2D-cultured 3T3-L1 cells. The 3T3-L1 cells were 2D-cultured under several sets of conditions: preadipocytes of 3T3-L1 cells (DIF−) and adipogenic-differentiated preadipocytes (DIF+) with or without 100 ng/mL IGF1 and/or 10 ng/mL M22 were subjected to a real-time analysis of metabolic function using a Seahorse XFe96 Extracellular Flux Analyzer. The oxygen consumption rate (OCR, panel (**A**)) and extra-cellular acidification rate (ECAR, panel (**B**)) were simultaneously measured at baseline and following injections of oligomycin (Oligo, complex V inhibitor), FCCP (a protonophore), rotenone/antimycin A (R/A, complex I/III inhibitors) and 2DG (a hexokinase inhibitor). Key parameters of mitochondrial respiration and glycolytic functions are shown in panel (**C**) and panel (**D**), respectively. Basal OCR was calculated by subtracting OCR with rotenone/antimycin A from the OCR at baseline. ATP-linked respiration was calculated by subtracting OCR with oligomycin from OCR at baseline. Maximal respiration was calculated by subtracting OCR with rotenone/antimycin A from OCR with FCCP. Basal ECAR was calculated by subtracting ECAR with 2DG from ECAR at baseline. Glycolytic capacity was calculated by subtracting ECAR with 2DG from ECAR with oligomycin. Energy map for basal OCR and ECAR is shown in panel (**E**). All experiments were performed in duplicate using fresh preparations (total *n* = 6), each of which consisted of a confluent 6-well dish. * *p* < 0.05.

**Figure 4 ijms-24-01110-f004:**
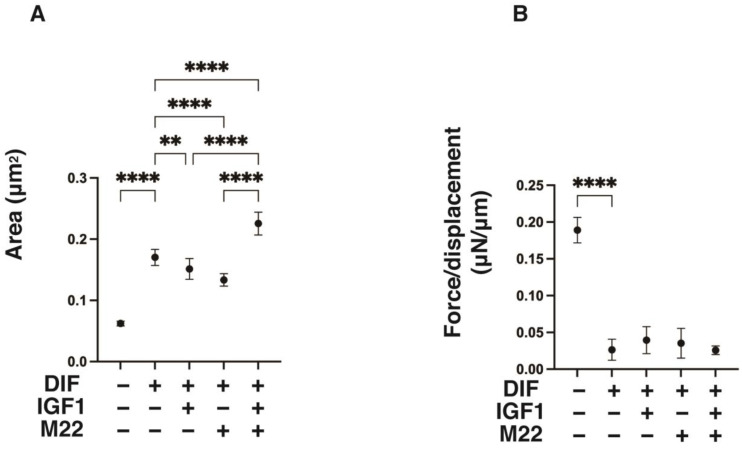
Effects of IGF1 and/or M22 on mean area sizes (**A**) and physical stiffness (**B**) of the 3T3-L1 3D spheroids. Three-dimensional spheroids of 3T3-L1 cells were prepared under several sets of conditions: preadipocytes of 3T3-L1 cells (DIF-) or adipogenic-differentiated preadipocytes (DIF+) with or without 100 ng/mL IGF1 and/or 10 ng/mL M22. The mean area sizes (μm^2^) of the 3D spheroids on day 7 were measured and plotted in panel (**A**). On day 7, a single 3D spheroid was placed on a 3 mm × 3 mm plate which was then compressed to 50% deformation during a period of 20 s; the required force (μN) was measured, and force/displacement (μN/μm) was plotted in panel (**B**). All experiments were performed in duplicate using fresh preparations (total *n* = 7–16 spheroids) each. ** *p* < 0.01, **** *p* < 0.001.

**Figure 5 ijms-24-01110-f005:**
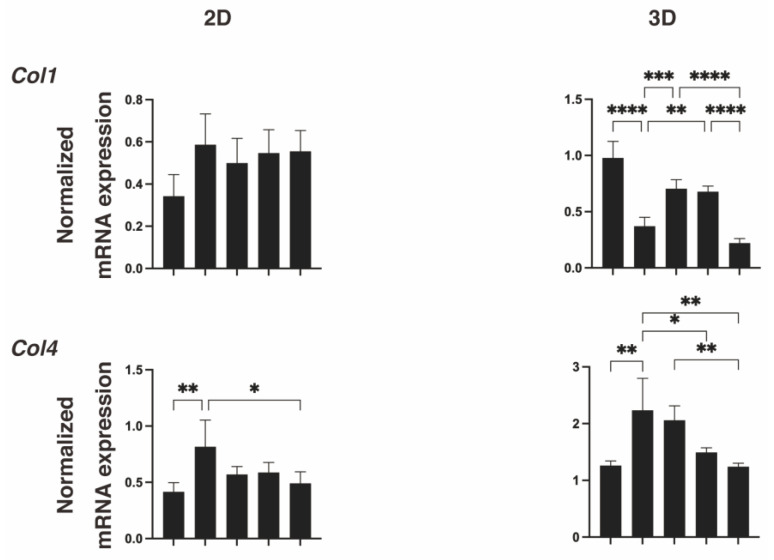
Effects of IGF1 and/or M22 on mRNA expressions of ECM molecules of 2D- or 3D-cultured 3T3-L1 cells. Two-dimensional or three-dimensional cultures of 3T3-L1 cells were prepared under several conditions: preadipocytes of 3T3-L1 cells (DIF-) or adipogenic-differentiated preadipocytes (DIF+) with or without 100 ng/mL IGF1 and/or 10 ng/mL M22. The specimens were subjected to qPCR analysis to estimate mRNA expression of major ECM molecules (*Col1*: collagen 1, *Col4*: collagen 4, *Col6*: collagen 6, *Fn*: fibronectin). All experiments were performed in duplicate using fresh preparations (total *n* = 4), each of which consisted of a confluent 6-well dish (2D) or 16 spheroids (3D). * *p* < 0.05, ** *p* < 0.01, *** *p* < 0.005, **** *p* < 0.001.

**Figure 6 ijms-24-01110-f006:**
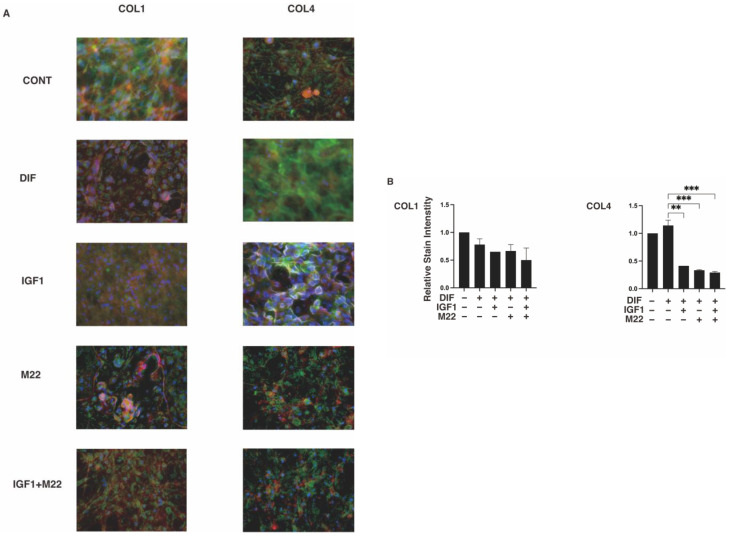
Representative confocal images showing the expression of COL1 and COL4 in 2D or 3D 3T3-L1 cells under several sets of conditions. On day 7, 2D cells and 3D cultures of spheroids of 3T3-L1 preadipocytes as the control (CONT) and adipogenic-differentiated preadipocytes in the absence (DIF) or presence of 100 ng/mL IGF1 and/or 10 ng/mL M22 were immunostained with specific antibodies against ECM molecules including collagen 1 (COL 1) and collagen 4 (COL 4), designated by a green color. Representative immunolabeling images are shown in panels (**A**) (2D) and (**C**) (3D) (scale bar: 100 μm) and their staining intensities (gray/pixel) were plotted in panels (**B**) (2D) and (**D**) (3D). All experiments were performed in duplicate each using fresh preparations (total *n* = 4). ** *p* < 0.01, *** *p* < 0.005.

## Data Availability

Not applicable.

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
