# Peer review of "Unexpected Crosslinking Effects of a Human Thyroid Stimulating Monoclonal Autoantibody, M22, with IGF1 on Adipogenesis in 3T3L-1 Cells"

_ijms, 2023, doi:10.3390/ijms24021110_

Round 1

Reviewer 1 Report

This study just revealed the effects of M22 and/or IGF1 on adipogenesis of 2D and 3D cultured 3T3-L1 preadipocytes, but the possible mechanism of this unexpected crosslinking effects has not been studied; Furthermore, the gene expression changes were detected by RT-PCT, Other methods are required, such as WB; 2D and 3D cell cultures were used in this study, and it is necessary to explain why 2D and 3D culture techniques were selected and the comparative analysis of their results. 

Author Response

Dear Editor,

Thank you very much for the constructive comments concerning our manuscript, " Unexpected crosslinking effects of human thyroid stimulating monoclonal autoantibody, M22, with IGF1 on adipogenesis in 3T3L-1 cells”. We examined the reviewers’ comments carefully and prepared a revised version of our paper that takes these comments into account for resubmission. Therefore, we will greatly appreciate it if you will consider our revised paper for possible publication in IJMS. The changes are listed below. In addition, during the course of this revision of this study, contributions of each author were greatly changed, and therefore, author contributions and their order were changed.

Reviewer 1

This study just revealed the effects of M22 and/or IGF1 on adipogenesis of 2D and 3D cultured 3T3-L1 preadipocytes, but the possible mechanism of this unexpected crosslinking effects has not been studied; Furthermore, the gene expression changes were detected by RT-PCT, Other methods are required, such as WB;

Answer; Thank you for these comments. As suggested, we completely agree that other methods such as WB should support gene expression data by qPCR. Nevertheless, especially in the case of 3D spheroids which contain only small amounts of proteins, in addition to abundant amounts of lipids, in order to obtain better resolutions of WB, at least 30-50 3D spheroids would be needed for one analysis and the obtained data would represent ambiguous mean levels of them, even if experiments were repeated. Therefore, instead of WB, we usually evaluate the protein expression levels of target molecules (ECMs) using immune-fluorescein labeling. In the case of this methodology, in each condition, the staining intensities of each 3D spheroid (n=4-8) could be subjected to statistical analysis. Therefore, we believe that immunostaining analysis of the specific molecules within a single 3D spheroid should show a better resolution as compared with WB. Similar to this 3D spheroid, 2D cells also contain large amounts of lipids and this may also affect the results of a WB analysis. Therefore, to study the expressions of ECM proteins, we performed immunocytochemistry rather than WB. Moreover, our immunolabeling intensities of COL1 and COL4 were in fact, quite similar results obtained by qPCR analysis (new Fig. 6 B and D). We, however, mistakenly missed including the staining intensity measurements data of the immunocyte-chemistry of ECMs (COL1 and COL4) in which there were significant difference among treatment groups, and therefore these data are now included within Fig. 6. In addition, this information is included in the 2nd last paragraph of Results; “To study these diverse effects by IGF-1 and/or M22 between 2D and 3D cell culture systems further, we evaluated the mRNA expressions of the major ECM proteins, COL1, 4 and 6, and FN. As shown in Fig. 5, upon DIF+, significant down-regulations or up-regulations of Col1 and Fn (3D), or Col4 and 6 (2D and 3D) were observed, consistent with our previous studies [24-26]. In terms of the effects of IGF-1 and/or M22, the mRNA expression of Col4 was significantly downregulated by M22 (2D and 3D) but those of Col1 were differently regulated by monotreatment (up-regulation) or combined treatment (down-regulation) (3D) (Fig. 5). Such differences in the COL1 and COL4 expressions among treatment groups were also confirmed by immunolabeling of 2D and 3D cultured 3T3-L1 cells (Fig. 6). Taking these collective data into account, we speculated that some unexpected simultaneous TSHR independent effects of IGF-1 and M22 may be exerted and that these could be more clearly detected in the case of the 3D spheroid culture system.”. 

2D and 3D cell cultures were used in this study, and it is necessary to explain why 2D and 3D culture techniques were selected and the comparative analysis of their results. 

Answer; Thank you for these comments. In terms of why we used 2D and 3D cell cultures in the present study, our previous studies using not only 3T3-L1 cells, but also cells from other sources demonstrated that 3D spheroid cultures reflect different biological aspects of these cells, and in fact, we were able to evaluate the physical properties, sizes and stiffness of a single living 3D spheroid that were not evaluated by a 2D cell culture system. In addition, even though same analyses including adipogenesis efficacies and the levels of expression of several factors and proteins, we observed difference between 2D and 3D cell cultures. Therefore, based upon these observations, we also believe that 3D spheroid cultures can reveal currently unidentified biological aspects that cannot be obtained using the 2D cell culture method. Thus, this information is included in the last 2 paragraphs of the Introduction; “In terms of physiology the spreading and differentiation of adipocytes should take place within a three-dimensional (3D) space. The use of a 3D cell culture system would be more desirable compared to a two-dimension (2D) planar cell culture method in adipocyte-related research. Although the use of such 3D cell culture systems has not become popular because of technical difficulties associated with this methodology, the use of such cultures has recently emerged as a useful strategy for modeling several human diseases [21]. In fact, it has been proposed that such a 3D cell culture technique would allow for more representative physiological cell-cell and cell-extracellular matrix (ECM) interactions than conventional 2D culture models [22]. Quite recently, our group independently succeeded in producing 3D spheroids using 3T3-L1 cells [23-28] as well as cells from other sources, including human orbital fibroblasts (HOFs) [29-31], human corneal stromal fibroblasts [32], human trabecular meshwork (HTM) [33-36] and human conjunctival fibroblasts (HconF) [37-40].     

Therefore, in this study, we report on an evaluation of the effects of M22 and/or IGF1 on several biological aspects of 2D and 3D cultured 3T3-L1 preadipocytes during their adipogenesis.”.

Reviewer 2 Report

In the current manuscript, the authors have explored the crosslinking effects of human thyroid stimulating monoclonal autoantibody, M22, with IGF1 on adipogenesis in 3T3L-1 cells. Authors utilized mouse adipocytes, two- or three-dimension culture of 3T3-L1 cells for the analysis. Authors performed lipid staining, a real-time cellular metabolic analysis, the mRNA expression of adipogenesis related genes and extracellular matrices (ECMs) molecules, and measurement of the size and physical properties of 3D spheroid.

The article is well structured into sections and subsections. English is clear and professional. It is within the scope of the journal. There are some minor comments to improve the article:

1)     Page 1, line 16-20: The sentence is too lengthy. Rephrasing is suggested to improve clarity.

2)     Page 2, line 101-102: Authors have mentioned “Information concerning the methods used in this study are shown in Supplemental methods”. However, there are no Supplemental methods available. There are two supplementary figures but no methods.

3)     Page 3, line 112 and 122: Similar issue, authors have mentioned “Information concerning the methods used in this study are shown in Supplemental methods”. However, there are no Supplemental methods available.

4)     Page 6, line 186, Figure 3: The labels of panel A and B are not legible, and the image resolution needs improvement.

5)     Page 6-10, Figure 3-6: The font size of axis and the labels vary a lot between figures.

6)     Page 7, Figure4: Figure 4 is placed in between the legend of Figure 3, which needs to be corrected.

7)     Page 13, line 364: The page numbers are missing for the reference.

8)     Supplementary, Figure 2: The figure labels are not legible, and the resolution of the image needs improvement.

Author Response

  1. Page 1, line 16-20: The sentence is too lengthy. Rephrasing is suggested to improve clarity.

Answer; Thank you for this comment. As suggested, this long sentence was changed; “To study the effects of the cross linking of IGF1 and/or the human thyroid stimulating monoclonal autoantibody (TSmAb), M22 on mouse adipocytes, two- and three-dimension (2D or 3D) culture of 3T3-L1 cells were prepared. Each sample was then subjected to following analyses; 1) lipid staining, 2) a real-time cellular metabolic analysis, 3) the mRNA expression of adipogenesis related genes and extracellular matrixes (ECMs) molecules including collagen (Col) -1, -4 and -6, and fibronectin (Fn) and 4) measurement of the size and physical properties of the 3D spheroids by a micro-squeezer.”.

  1. Page 2, line 101-102: Authors have mentioned “Information concerning the methods used in this study are shown in Supplemental methods”. However, there are no Supplemental methods available. There are two supplementary figures but no methods.

Answer; Thank you for this comment and I apologize for this careless mistake in not attaching the supplemental method file. Therefore, the corresponding material is now included.

  1. Page 3, line 112 and 122: Similar issue, authors have mentioned “Information concerning the methods used in this study are shown in Supplemental methods”. However, there are no Supplemental methods available.

Answer; Thank you for this comment. We apologize for this careless mistake not to attach supplemental method file. Therefore, the corresponding material is now included.

  1. Page 6, line 186, Figure 3: The labels of panel A and B are not legible, and the image resolution needs improvement.

Answer; Thank you for this comment. As suggested, the resolution of the image in Figure 3 was improved.

  1. Page 6-10, Figure 3-6: The font size of axis and the labels vary a lot between figures.

Answer; Thank you for this comment. As pointed out, the font size of the axis and the labels of Fig. 3-6 were unified.

  1. Page 7, Figure4: Figure 4 is placed in between the legend of Figure 3, which needs to be corrected.

Answer; Thank you for this comment. As pointed out, Fig. 4 was moved properly.

  1. Page 13, line 364: The page numbers are missing for the reference.

Answer; Thank you for this comment. As pointed out, page numbers of the corresponding ref are now included.

  1. Supplementary, Figure 2: The figure labels are not legible, and the resolution of the image needs improvement.

Answer; Thank you for this comment. As suggested, the resolution for the image in supplementary Figure 2 was improved.

Round 2

Reviewer 1 Report

No further comments.